# Task Success Prediction for Open-Vocabulary Manipulation Based on Multi-Level Aligned Representations

**Miyu Goko\*  Motonari Kambara\*  Daichi Saito  Seitaro Otsuki  Komei Sugiura**

Keio University, Japan

{miyu.goko, motonari.k714, daichi-s, otsu8sei14, komei.sugiura}@keio.jp

**Abstract:** In this study, we consider the problem of predicting task success for open-vocabulary manipulation by a manipulator, based on instruction sentences and egocentric images before and after manipulation. Conventional approaches, including multimodal large language models (MLLMs), often fail to appropriately understand detailed characteristics of objects and/or subtle changes in the position of objects. We propose Contrastive $\lambda$-Repformer, which predicts task success for table-top manipulation tasks by aligning images with instruction sentences. Our method integrates the following three key types of features into a multi-level aligned representation: features that preserve local image information; features aligned with natural language; and features structured through natural language. This allows the model to focus on important changes by looking at the differences in the representation between two images. We evaluate Contrastive $\lambda$-Repformer on a dataset based on a large-scale standard dataset, the RT-1 dataset, and on a physical robot platform. The results show that our approach outperformed existing approaches including MLLMs. Our best model achieved an improvement of 8.66 points in accuracy compared to the representative MLLM-based model.

**Keywords:** Task Success Prediction, Open-Vocabulary Manipulation, Multi-Level Aligned Visual Representation

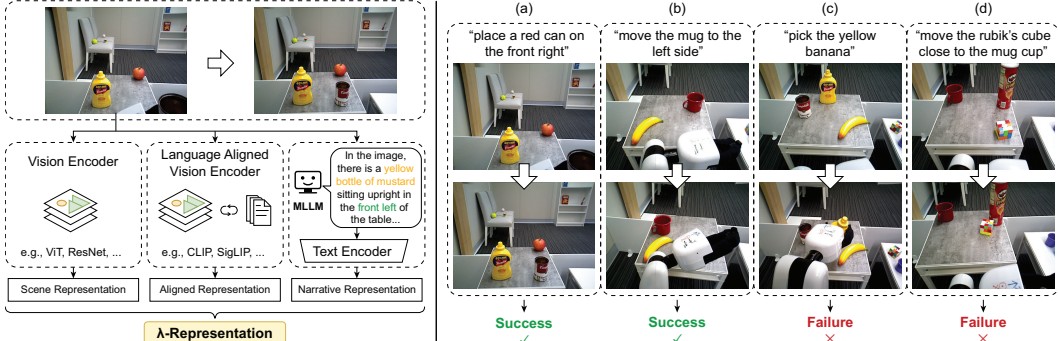

Figure 1: (left) An overview of the novel representation: $\lambda$-Representation, which is an integration of three types of representations. (right) A few examples of our task. The task is to predict success or failure based on an open-vocabulary instruction sentence, and egocentric images taken before and after the manipulation.

## 1 Introduction

Task success prediction in object manipulation ensures precise and efficient operations, enhancing reliability and consistency across robotic applications in healthcare, manufacturing, agriculture, and logistics. For example, in object manipulation tasks such as assembling parts in manufacturing [1, 2] and harvesting crops in agriculture [3, 4], task success prediction can improve the quality, efficiency, and productivity of the tasks. The ability of a manipulator to accurately predict the success or failure

---

\* denotes equal contribution.

Project page available at https://5ei74r0.github.io/contrastive-lambda-repformer.page/

of sub-tasks is particularly important for long-horizon tasks because a failure in a sub-task may affect subsequent ones.

In this study, we focus on a task which involves predicting the success or failure of an open-vocabulary manipulation, given an instruction sentence and egocentric images before and after the manipulation. Typical use cases involve a scene where a manipulator is given the instruction sentence: "Place the mug into the sink." In the case where the manipulator drops the mug, the model is expected to predict failure based on the instruction and egocentric images. On the other hand, in the case where the mug is successfully placed into the sink, the model is expected to predict success.

Our target task is challenging because it demands two key aspects. First, it is necessary to have an adequate understanding of the changes in the images taken pre- and post-manipulation, information about the objects in the images, and open-vocabulary instructions. The task also requires the model to determine if the elements above align. Even multimodal large language models (MLLMs [5, 6, 7]) demonstrate limited performance on this task as we will show in the experimental results (See Section 4.2). This is because MLLMs often fail to appropriately understand detailed characteristics of objects (e.g., colors and shapes) and subtle changes in the position of objects, both of which are critical for success prediction.

We propose Contrastive $\lambda$-Repformer, which performs task success prediction for table-top open-vocabulary manipulation by aligning images with instruction sentences. The method achieves this by utilizing visual representations that integrate three key types of features which are the following: (i) features that preserve local image information, (ii) features aligned with natural language, and (iii) features structured through natural language (Fig. 1). This addresses a problem in conventional methods that rely solely on a single visual representation extraction mechanism: they struggle to extract both detailed visual features, such as textures and shapes of objects, and global structural representations, such as spatial relationships between objects. The method also employs a representation of the difference between the images, allowing it to effectively align the manipulations with the instruction sentences. This alignment enables the model to understand instruction sentences by considering the specific characteristics of objects and their spatial relationships.

We make the following contributions:

- We introduce $\lambda$-Representation Encoder, which computes the aforementioned three types of visual representations and integrates them into $\lambda$-Representation for the image. $\lambda$-Representation integrates three types of features: (i) features retaining visual characteristics such as colors and shapes, (ii) features aligned with natural language, and (iii) features that are structured through natural language.
- We propose Contrastive $\lambda$-Representation Decoder, which identifies the difference between $\lambda$-Representations of two images. This allows the model to take into consideration the alignment between the differences in the images and the instruction sentence when performing task success prediction.

## 2   Related Work

Recent research on foundation models (e.g. [8, 5, 9]) has made significant breakthroughs in the field of robotics [10, 11, 12, 13, 14, 15]. Several surveys [16, 17, 18] provide a comprehensive summary of various MLLM-based models in the robotics field. In multimodal language understanding tasks for robotics, various datasets are utilized as representative benchmarks in real-world settings [19, 20, 21, 22] and in simulation settings [23, 24, 25, 22]. These datasets primarily focus on object manipulation tasks within indoor environments.

**LLM-Based Task Planning.**  For object manipulation tasks, large language models (LLMs) are often employed as task planners [26, 27, 28, 13, 29, 30, 31]. For example, in some studies, LLMs are utilized to generate sub-goals from high-level instruction sentences [26, 27, 28, 31]. This approach involves replanning using the LLMs based on feedback received from the environment when a task failure is detected. On the other hand, some methods (e.g. [13, 29]) use LLMs to directly generate Python code for robot policies based on natural language instructions. Other works have

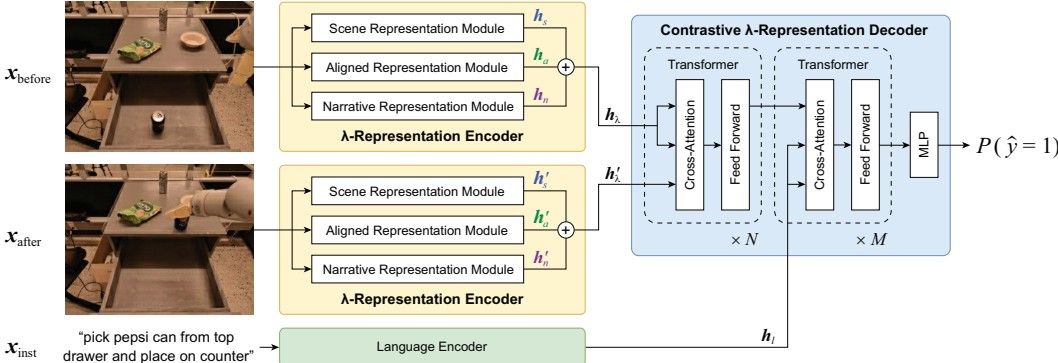

Figure 2: Overview of Contrastive $\lambda$-Repformer. Given an instruction sentence and images before and after manipulation, our model outputs the predicted probability that the robot successfully performed the manipulation.

also explored LLM-based reward generation, including grounding the reward in the 3D observation space [10, 32, 33, 34, 35]. While REFLECT [30] is closely related to our method, it determines task success by predefining the target state for each object class and verifying whether these states are achieved. Consequently, unlike our method, it is difficult for REFLECT to perform success prediction without using predetermined target states.

Our method is also closely related to MLLM-based task planning models (e.g. [27, 28, 36]). Unlike them, we employ MLLMs for the purpose of structuring images through natural language. Furthermore, we introduce a mechanism that extracts visual representations through two other types of modules and integrates them alongside the MLLMs. This allows the model to consider visual representation with multi-level alignment that simple MLLM-based approaches cannot fully capture.

**Task Success Prediction.** Most reward-based approaches require both expert knowledge and significant effort to manually design rewards that consider all of the states during manipulation. Meanwhile, our method needs only the states before and after manipulation. Inverse reinforcement learning methods [37, 38, 39, 40] aim to learn reward functions from optimal demonstrations. However, obtaining such demonstrations can be costly and sometimes unfeasible. Alternatively, some strategies train agents by acquiring rewards through interactive human feedback [41, 42, 43, 44]. However, this approach is limited by the necessity of having human supervision for real-time queries. In contrast, representative methods [26, 45] that do not require optimal demonstrations or human supervision have been proposed. Notably, PaLM-E [26] is one prominent object manipulation model that utilizes Visual Question Answering, achieving a 91% success rate on the failure detection task with the dataset proposed in [27]. However, PaLM-E has a large model size, which is a problem in robotics where computational resources are often limited. Also, the dataset used for failure detection in [26] included only 101 episodes and 15 objects. Thus, we constructed a new dataset based on the RT-1 dataset [19], which has approximately 1,000 episodes and 30 objects.

The collision prediction task during object manipulation is also related to our task. For example, there are some post-collision decision strategies (e.g. [46]). Furthermore, several methods predict collisions from an image and a placement policy [47, 48, 49]. Our method differs from these in that it can take into account factors other than collisions that contribute to task failure.

**Using Captions.** Scene change captioning models aim to generate descriptions about the differences between two images [50, 51, 52, 53, 54, 55]. This task is related to our task in that it requires the identification of the differences between two images. However, they have difficulty handling instructional sentences as input. Thus, it is not possible to directly apply those models to our task.

## 3 Proposed Method

Our target problem is to predict whether an open-vocabulary manipulation task was performed successfully, given an instruction sentence and egocentric images taken before and after the manipulation. We define this task as Success Prediction for Open-vocabulary Manipulation (SPOM). In this

task, models are expected to appropriately predict the success or failure of an object manipulation. The inputs consist of an instruction sentence, one egocentric image taken before the manipulation, and another taken after. The expected output is the predicted probability $P(\hat{y} = 1)$, indicating the probability that the manipulator successfully executed the open-vocabulary manipulation specified in the instruction sentence. Here, $\hat{y}$ represents the success or failure of the manipulation, with '1' indicating success. In this study, we only use egocentric images as input images. Note that in some images, the scene is partially occluded by the manipulator. While the task is feasible, this often makes it challenging, as the objects or areas may be partially occluded.

Fig. 2 shows the structure of the proposed method, Contrastive $\lambda$-Repformer. Its input is defined as $x = \{x_{\mathrm{inst}}, x_{\mathrm{before}}, x_{\mathrm{after}}\}$, where $x_{\mathrm{inst}}$ represents a tokenized instruction, while $x_{\mathrm{before}}$ and $x_{\mathrm{after}}$ represent RGB images taken before and after manipulation, respectively. The main modules of the proposed method are $\lambda$-Representation Encoder and Contrastive $\lambda$-Representation Decoder.

### 3.1 $\lambda$-Representation

In existing Vision-and-Language studies, there are primarily three approaches for extracting visual features. The first approach uses unimodal image encoders [56, 57, 58] to extract visual features like textures and edges; we refer to these features as "Scene Representation." The second approach employs multimodal image encoders [8, 59, 60, 61] to extract visual features aligned with natural language, referred to here as "Aligned Representation." The third approach utilizes MLLMs [5, 6, 7] to extract structural features that directly represent complex referring expressions and spatial relationships through natural language, termed "Narrative Representation" in this paper.

However, most existing methods do not comprehensively handle all the above representations, limiting the expressiveness of visual features. Specifically, Scene Representation, despite its ability to capture visual information like shapes and colors from images, cannot extract complex referring relations, including spatial relations. This limitation highlights the insufficiency of using this representation exclusively. In addition, while Narrative Representation is capable of extracting structural features through natural language, it is difficult to capture all the detailed visual features, such as textures, with only this representation. Unlike these representations, Aligned Representation is aligned with natural language, sharing characteristics with both Scene and Narrative Representations. However, using only Aligned Representation often leads to a lack of ability to structurally understand complex referring expressions in instruction sentences, because it does not extract structural features through natural language. From the above, it is expected that we can obtain sufficient visual representations by using all these features in parallel.

### 3.2 $\lambda$-Representation Encoder

We introduce $\lambda$-Representation Encoder, designed to generate $\lambda$-Representation effectively. In this module, we obtain the three types of visual representations and integrate them into $\lambda$-Representation. As shown in Fig. 2, this module consists of three sub-modules: Scene Representation Module, Aligned Representation Module, and Narrative Representation Module. $\lambda$-Representation Encoder takes either $x_{\mathrm{before}}$ or $x_{\mathrm{after}}$ as input. The following explanation will focus solely on $x_{\mathrm{before}}$, because the same process is applied to $x_{\mathrm{after}}$.

First, we obtain Scene Representation $h_s = f_{\mathrm{srm}}(x_{\mathrm{before}})$, where $f_{\mathrm{srm}}(\cdot)$ represents Scene Representation Module. Scene Representation Module consists of several backbone networks. In this paper, we use ViT [56], DINOv2 [58], and the CLIP image encoder [8] as backbone networks. For ViT and DINOv2, the output features are used, while the intermediate features are utilized for the CLIP image encoder. Then, $h_s$ is acquired by concatenating them.

Next, we acquire Aligned Representation $h_a$ using Aligned Representation Module, which is composed of multimodal foundation models. These features can be regarded as Aligned Representations, because they are well-aligned with natural language. We employ the CLIP image encoder and extract its output features.

Subsequently, Narrative Representation $h_n$ is obtained using Narrative Representation Module, containing a MLLM and multiple text embedders. We utilize InstructBLIP [7] to generate a description

from $x_{\text{before}}$. We designed a text prompt to focus on the colors, sizes, and shapes of objects, as well as how they are placed, their positions within the image, and their relative positions to other objects. From the output of InstructBLIP, we acquire its features using BERT and text-embedding-3-large [62]. Then, these features are concatenated to obtain $h_n$. Finally, we obtain $\lambda$-Representation for $x_{\text{before}}$, denoted as $h_\lambda = \left[h_s^\top, h_a^\top, h_n^\top\right]^\top$. Similarly, we obtain $h_\lambda'$ as $\lambda$-Representation for $x_{\text{after}}$.

### 3.3 Contrastive $\lambda$-Representation Decoder

We introduce Contrastive $\lambda$-Representation Decoder to create a representation of the difference between $h_\lambda$ and $h_\lambda'$. Since the effects of the manipulation are included in the change between the images, the representation allows the model to focus on the difference, which may be attributed to the manipulation. On the other hand, a difference between the images does not necessarily indicate the success of the task specified by the given instruction sentence. For example, in the case shown in Fig. 2, if the Pepsi can were to fall over, there would be a difference between the two images; however, the manipulation should be considered a failure. Thus, it is hard to consider the success of a manipulation based solely on the differences between images. Consequently, when predicting the success or failure of a manipulation, it is important to consider the alignment between the difference representation and the instruction sentence.

The inputs of this module are $h_\lambda$, $h_\lambda'$, and $h_l$, and the output is $P(\hat{y} = 1)$. First, the representation of the difference $h_{\text{diff}}$ between the two images are obtained as follows:

$$h_{\text{diff}} = \text{CrossAttn}\left(h_\lambda', h_\lambda\right),\tag{1}$$

where $\text{CrossAttn}\left(\cdot, \cdot\right)$ represents the cross-attention operation. We define this operation using two arbitrary matrices $X_A$ and $X_B$ as follows:

$$\text{CrossAttn}\left(X_A, X_B\right) = \text{softmax}\left(\frac{X_A W_q (X_B W_k)^\top}{\sqrt{d_k}}\right) X_B W_v,\tag{2}$$

where $W_q$, $W_k$, and $W_v$ are trainable weights, and $d_k$ denotes a dimension of $X_B W_k$. Then, the alignment feature $h_{\text{align}}$ between $h_{\text{diff}}$ and $h_l$ is computed as follows:

$$h_{\text{align}} = \text{CrossAttn}\left(h_{\text{diff}}, h_l\right).\tag{3}$$

Finally, we compute $P(\hat{y} = 1)$ from $h_{\text{align}}$ as the output of this module using a multi-layer perceptron. We use the cross entropy loss as the loss function.

## 4 Experimental Results

### 4.1 Experimental Setup

We constructed the novel SP-RT-1 dataset from the RT-1 dataset for the SPOM task. The task requires all of the following components for each episode: an instruction sentence, images taken before and after the manipulation, and labels indicating the success or failure of the manipulation. The RT-1 dataset is a standard, large-scale dataset for real-world open-vocabulary manipulation. It includes instruction sentences, images collected during manipulation, and binary rewards. Because the RT-1 dataset cannot be utilized directly, the SP-RT-1 dataset was assembled from the RT-1 dataset. We collected the first and last images of each episode and got the ground truth success/failure labels by using the binary rewards from the RT-1 dataset. The dataset was preprocessed by modifying the instruction sentences. Data cleansing was conducted because the rewards were sometimes erroneous. The details of the SP-RT-1 dataset are explained in Section A.3.1.

We used UNITER-base/large [59], the method by Xiao et al. [45], InstructBLIP Vicuna-7B (InstructBLIP) [7], GPT-4 Turbo with Vision (GPT-4V) [5], and Gemini 1.0 Pro Vision (Gemini) [6] as baseline methods. The capability of InstructBLIP was evaluated in a zero-shot manner, while GPT-4V and Gemini were evaluated in both zero-shot and few-shot settings. Each method was used as a baseline method for the following reasons. UNITER demonstrated competitive performance in many Vision-and-Language tasks, including Visual Question Answering tasks. The model by Xiao et al. is a failure detection model based on the two images and an instruction sentence. This performance is reported to be competitive with PaLM-E, a large-scale model for object manipulation

in robotics. Additionally, InstructBLIP, GPT-4V, and Gemini are representative MLLMs that have been pretrained on large-scale datasets and have demonstrated outstanding performance on various tasks. The details of baseline methods are explained in Section A.3.4.

For a comprehensive evaluation, we also validated our model in a physical environment using a mobile manipulator with zero-shot settings (SP-HSR benchmark). Fig. 3 shows the experimental environment, which is based on the standardized environment of WRS2020 [63]. We used Toyota's Human Support Robot, which is standardized in RoboCup@Home competitions [64]. This dataset was annotated by humans during its construction. Specifically, each sample was labeled as 'Success' if the images matched the instructions; otherwise, it was labeled as 'Failure.' In the experiment, all methods were evaluated in zero-shot settings. This means no additional training was conducted using the collected data. The details of the dataset for this experiment are explained in Section A.3.2. The implementation details are also explained in Section A.3.3.

Figure 3: Experimental environment. The left and right images show the state before and after manipulation, respectively. Instruction sentences, such as "place a mug in front of the banana," were created based on the situation before the manipulation. Examples of the egocentric images are shown at the top right of each exocentric image.

### 4.2 Quantitative Results

Table 1 presents the quantitative results of a comparison between several baseline methods and Contrastive $\lambda$-Repformer. As listed in Table 1, on the SP-RT-1 dataset, Contrastive $\lambda$-Repformer achieved the highest accuracy of 80.80%, outperforming UNITER-base, UNITER-large, and the method by Xiao et al. with accuracies of 62.78%, 63.52%, and 68.26%, respectively. Furthermore, Contrastive $\lambda$-Repformer also outperformed MLLMs: InstructBLIP, GPT-4V (Zero-shot), GPT-4V (Few-shot), Gemini (Zero-shot), and Gemini (Few-shot) with accuracies of

| Method | Accuracy [%] | |
| | SP-RT-1 | SP-HSR |
| --- | --- | --- |
| UNITER-base [59] | $62.78 \pm 1.01$ | $52 \pm 1.6$ |
| UNITER-large [59] | $63.52 \pm 1.84$ | $48 \pm 1.8$ |
| Xiao et al. [45] | $71.59 \pm 1.95$ | - |
| InstructBLIP [7] | $50.50 \pm 0.00$ | $50 \pm 0.0$ |
| GPT-4V [5] (Zero-shot) | $63.90 \pm 1.04$ | $59 \pm 1.9$ |
| GPT-4V [5] (Few-shot) | $72.14 \pm 0.92$ | $56 \pm 1.9$ |
| Gemini [6] (Zero-shot) | $67.28 \pm 0.80$ | $53 \pm 0.40$ |
| Gemini [6] (Few-shot) | $68.44 \pm 0.76$ | $53 \pm 3.3$ |
| Contrastive $\lambda$-Repformer | $\textbf{80.80} \pm 0.86$ | $\textbf{60} \pm 1.8$ |
| Human (Reference) | 90 | 79 |

Table 1: Quantitative results of the baseline and proposed methods on the SP-RT-1 dataset and the SP-HSR benchmark. Here, SP-HSR represents our benchmark using a physical environment. Bold indicates the accuracy with the highest value.

50.50%, 63.90%, 72.14%, 67.28%, and 68.44%, respectively. These results demonstrate that the proposed method outperformed both zero/few-shot MLLMs and other baseline methods. The differences in accuracy between Contrastive $\lambda$-Repformer and each baseline method were statistically significant ($p < 0.001$).

Table 1 also shows the quantitative results of the SP-HSR benchmark. The accuracies of UNITER-base, UNITER-large, and Contrastive $\lambda$-Repformer were 52%, 48%, and 60%, respectively. Moreover, InstructBLIP, GPT-4V (Zero-shot), GPT-4V (Few-shot), Gemini (Zero-shot), and Gemini (Few-shot) were 50%, 59%, 56%, 53%, and 53%, respectively. The accuracies of most methods were nearly at chance level. On the other hand, GPT-4V (Zero/Few-shot) and Contrastive $\lambda$-Repformer showed better results compared to the other methods. Furthermore, the accuracy of Contrastive $\lambda$-Repformer slightly outperformed GPT-4V (Zero-shot) and (Few-shot).

We conducted a subject experiment with five subjects to evaluate the human performance for the task. For the SP-RT-1 dataset, 100 samples were randomly selected from the test set and the subjects performed the SPOM task on these samples, achieving an accuracy of 90%. For the SP-HSR benchmark, the entire dataset was used, with humans achieving an accuracy of 79%. From this result, it is found that the SPOM task can be difficult even for humans.

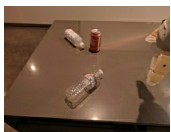 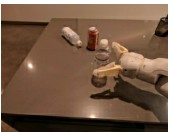 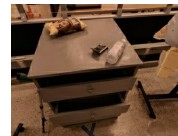 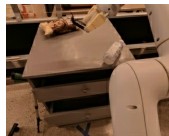 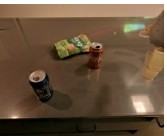 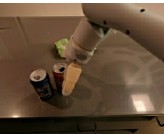

(i) "place water bottle upright"      (ii) "pick rxbar chocolate"      (iii) "pick apple from white bowl"

Figure 4: Successful cases of Contrastive $\lambda$-Repformer on the SP-RT-1 dataset. Examples (i) and (ii) are true positive cases, and (iii) is a true negative case. In each example, the left and right images show the scene before and after the manipulation, respectively.

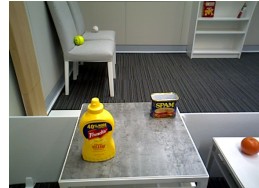 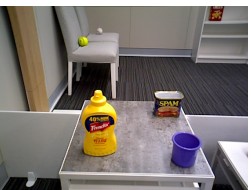 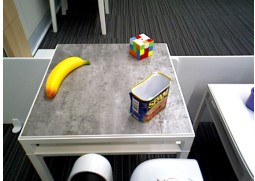 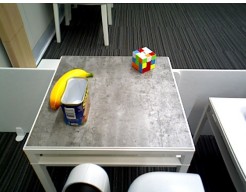

(i) "place a purple cup on the front right"      (ii) "move the rubik's cube close to the banana"

Figure 5: Qualitative results of the proposed method in zero-shot transfer experiment. Examples (i) and (ii) are true positive and true negative cases, respectively. In each example, the left and right images show the scene before and after the manipulation, respectively.

## 4.3 Qualitative Results

Fig. 4 exhibits successful cases of Contrastive $\lambda$-Repformer on the SP-RT-1 dataset. Fig. 4 (i) and (ii) are true positive cases, and Fig. 4 (iii) is a true negative case. Fig. 4 (i) presents an example where the given instruction was "place water bottle upright." The manipulator successfully manipulated the water bottle, setting it down so that it was upright. Therefore, the example was labeled as a success. Contrastive $\lambda$-Repformer correctly predicted success for this example where all of the baseline methods excluding InstructBLIP failed to do so. Fig. 4 (ii) is an instance where the manipulator executed the instruction of "pick rxbar chocolate," which can be observed from the fact that the chocolate is being held by the manipulator in the right image. While most of the baseline methods predicted that the example was a failure, Contrastive $\lambda$-Repformer was able to predict it as a success. Fig. 4 (iii) is one example where the manipulator was not able to follow the instruction: "pick apple from white bowl." Neither the apple nor the white bowl is visible in either of the images, which indicates a failure in the manipulation. Contrastive $\lambda$-Repformer successfully predicted that the manipulator failed in the task. Meanwhile, all of the baseline methods predicted success.

Fig. 5 shows successful examples in the SP-HSR benchmark. Fig. 5 (i) and (ii) are true positive and true negative cases, respectively. In Fig. 5 (i), the instruction given was "place a purple cup on the front right." The manipulator successfully put a purple cup on the front right of the table. Therefore, this episode was labeled as a success. Contrastive $\lambda$-Repformer successfully predicted it, while UNITER-base/large incorrectly predicted it as a failure. This result shows that the proposed method could appropriately understand the spatial expression 'front right.' Fig. 5 (ii) shows an episode in which the instruction "move the rubik's cube close to the banana" was given. This episode was labeled as a failure because the manipulator moved a blue can instead of the Rubik's cube. In this episode, Contrastive $\lambda$-Repformer made an appropriate prediction, while Gemini and InstructBLIP failed. This episode shows that the proposed method can also appropriately align natural language expressions with objects in the image.

## 4.4 Ablation Study

We conducted ablation studies to investigate the contribution of each representation in $\lambda$-Representation. Table 2 presents the results. We set the following conditions:

**Scene Representation Ablation.** We removed Scene Representation from $\lambda$-Representation to assess its contributions. From Table 2, it can be observed that the accuracy of Model (i) was 73.72%, which was 7.08 points lower than that of Model (vii). This signifies that Scene Representation enhanced the visual representation by capturing detailed visual information such as shapes and colors.

**Aligned Representation Ablation.** Aligned Representation was omitted from $\lambda$-Representation to analyze its contributions. As shown in Table 2, the accuracy of Model (ii) was 79.94%, which was

0.86 points lower than that of Model (vii). This shows that Aligned Representation improved the alignment between the instructions and the images, including better identification of object names.

**Narrative Representation Ablation.** We removed Narrative Representation from $\lambda$-Representation to investigate its contributions. Table 2 shows that Model (iii) achieved an accuracy of 79.70%, which was 1.10 points lower than that of Model (vii). This indicates that Narrative Representation enhanced the visual representation by extracting features structured through natural language.

| Model | SR | AR | NR | Accuracy [%] |
|-------|-----|-----|-----|--------------|
| (i)   |     | ✓   | ✓   | $73.72 \pm 0.86$ |
| (ii)  | ✓   |     | ✓   | $79.94 \pm 0.40$ |
| (iii) | ✓   | ✓   |     | $79.70 \pm 0.89$ |
| (iv)  | ✓   |     |     | $80.36 \pm 0.62$ |
| (v)   |     | ✓   |     | $74.90 \pm 0.55$ |
| (vi)  |     |     | ✓   | $61.80 \pm 0.47$ |
| (vii) | ✓   | ✓   | ✓   | $\mathbf{80.80} \pm 0.86$ |

Table 2: Results of ablation study. Bold indicates the highest value. SR, AR and NR represent Scene, Aligned and Narrative Representation, respectively.

The accuracy of the models with only Scene Representation, Aligned Representation, and Narrative Representation were 80.3, 74.0, and 61.8, respectively. From this result, it can be concluded that Scene Representation alone yields the highest accuracy when only a single representation is used, but has a lower accuracy than our proposed model with all three of the representations.

The results demonstrate that each representation in $\lambda$-Representation significantly contributed to the overall performance of the model. Particularly, it was found that Scene Representation contributed the most to performance improvement. Therefore, it can be said that this task is too challenging to be solved solely by MLLMs without explicitly using features of detailed characteristics. Constructing a model that integrates features obtained from MLLMs and other features, such as those represented by $\lambda$-Representation, is effective for the task.

## 5    Conclusions and Limitations

In this study, we focused on a task to predict the success or failure of open-vocabulary manipulation, given an instruction sentence and egocentric images before and after the manipulation. Our contributions can be emphasized as follows: We introduced the $\lambda$-Representation Encoder, which generates the multi-level aligned visual representation, $\lambda$-Representation. This representation consists of: (i) features that maintain visual characteristics such as colors and shapes, (ii) features aligned with natural language, and (iii) features structured through natural language. We also introduced Contrastive $\lambda$-Representation Decoder, which finds differences between two images, and enables the model to consider the alignment between the difference and an instruction sentence. Additionally, Contrastive $\lambda$-Repformer outperformed baseline methods, including representative MLLMs.

**Limitations.** Although Contrastive $\lambda$-Repformer generated compelling results, it has several limitations. Firstly, it assumes the availability of either local (e.g. InstructBLIP [7], LLaVA [65]) or cloud-based (e.g. Gemini [6], GPT-4V [5]) MLLMs to extract Narrative Representation; however, there are limitations associated with them. The former has limitations in terms of memory and inference time due to the large parameter size during inference. The latter cannot be used within a stand-alone system. Second of all, as stated in Section 3, the input images of this study were egocentric images. Thus, there were samples where objects directly related to the manipulation were occluded or were outside the photographed scene. In these cases, it is difficult to execute the task appropriately. Finally, in the experiments conducted for this study, we focused on a limited set of open-vocabulary manipulation tasks, such as pick and place. Therefore, Contrastive $\lambda$-Repformer is not intended to be applied directly to tasks such as navigation and mobile manipulation, making it difficult to solve such tasks. In future research, we plan to apply the method to a wide range of manipulation and navigation tasks (e.g., [66, 27, 14]). A possible solution could be to compare the images taken before and after the mobile manipulation. For example, when given an instruction "move the cup on the dining table to the shelf," a model can predict the success of the task based on the images of the dining table prior to the task and the shelf afterward.

**Acknowledgments**

This work was partially supported by JSPS KAKENHI Grant Number 23K03478, JST Moonshot, and NEDO.

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

# Appendix

## 1 Additional Related Work

Cap4Video [67] is a representative video retrieval model based on natural language queries. This method is similar to our proposed method in that it generates visual representations through natural language. However, Cap4Video uses only the features aligned with natural language, extracted by CLIP. It neither uses features that preserve local image information nor those structured through natural language. Thus, its capability to understand complex referring expressions is limited. In contrast, our method uses all three types of features. Additionally, Cap4Video requires human-annotated captions, while our method does not.

## 2 Details of Modules

Our method is primarily composed of three modules: $\lambda$-Representation Encoder, Contrastive $\lambda$-Representation Decoder, and Language Encoder. Below is a detailed explanation of Language Encoder.

We extract language feature $h_l$ from $x_{\text{inst}}$ using Language Encoder. In this module, we process $x_{\text{inst}}$ with BERT [68] to obtain the feature corresponding to the CLS token $l_{\text{BERT}}$. We also use the CLIP text encoder [8] and text-embedding-ada-002 [69] in parallel to extract the language features $l_{\text{CLIP}}$ and $l_{\text{ada}}$, respectively, from $x_{\text{inst}}$. Finally, we concatenate them to obtain the language feature $h_l = \left[ l_{\text{BERT}}^{\mathsf{T}}, l_{\text{CLIP}}^{\mathsf{T}}, l_{\text{ada}}^{\mathsf{T}} \right]^{\mathsf{T}}$.

## 3 Details of Experimental Setup

### 3.1 SP-RT-1 Dataset

As described in Section 4.1, we constructed the SP-RT-1 dataset from the RT-1 dataset [19] for our task. The details are described below. We collected the first and last images of each episode. The dataset was preprocessed by modifying the instruction sentences. In the RT-1 dataset, 43.6% of the negative samples were incorrectly labeled as negative, despite the manipulator having successfully executed the manipulation. We replaced the instruction sentences for the incorrectly annotated samples with alternative sentences that were randomly selected to create negative samples. This strategy was chosen instead of converting them to positive samples, because the original dataset contained fewer negative samples than positive samples, and converting negative samples to positive samples would further reduce the proportion of negative samples.

The SP-RT-1 dataset consisted of a total of 13,915 samples, with a vocabulary size of 49, a total word count of 78,790, and an average sentence length of 5.66. The dataset contains 10,000 positive samples and 3,915 negative samples. The SP-RT-1 dataset contained 11,915, 1,000, and 1,000 samples in the training, validation, and test sets, respectively. We used the training, validation, and test sets to estimate parameters, tune hyperparameters, and evaluate models, respectively. We computed the accuracy on the validation set every epoch. The performance on the test set was evaluated using the model that achieved the highest accuracy on the validation set.

**Other related datasets and benchmarks.** For multimodal language understanding tasks in robotics, various datasets and benchmarks are used in both real-world [20, 70, 71] and simulation [72, 73, 74, 75] settings. Among them, the RT-1 dataset is the most relevant to our target task of success prediction for object manipulation. Additionally, VLMbench [25] is a standard benchmark for object manipulation tasks on a tabletop. It provides natural language instructions, labels indicating the success or failure of each manipulation, and images captured from five camera views.

### 3.2 SP-HSR Benchmark

For a comprehensive evaluation, we validated the proposed method in a physical environment using a mobile manipulator with zero-shot transfer settings (SP-HSR benchmark). The data was collected in the environment described in Section 4.1. In this experiment, we used a subset of the YCB objects [76], which are standard objects for manipulation research. These selections were based on their suitability for grasping by the HSR end-effector.

In the experiment, we randomly selected up to four objects and arranged them on the table. Then, executable open-vocabulary instruction sentences were created and assigned to the episodes. The manipulations were performed by remote controlling the robot. The images of the scene before and after the manipulations were taken using the head-mounted camera of the robot. In total, 112 episodes were collected, with 56 episodes for both positive and negative samples.

### 3.3 Implementation Details

Table 3 shows the experimental settings for the proposed method. Our model had approximately 64M trainable parameters and 7.25G multiply-add operations. We trained our model on a GeForce RTX 4090 with 24 GB of GPU memory and an Intel Core i9-13900KF with 64 GB of RAM. It took approximately 1.5 hours to train our model on the SP-RT-1 dataset. The inference time was approximately 1.6 ms/sample.

| Optimizer | Adam ($\beta_1 = 0.9, \beta_2 = 0.999$) |
|---|---|
| Learning rate | $1.0 \times 10^{-6}$ |
| Weight decay | $1.0 \times 10^{-1}$ |
| Batch size | 32 |
| Epoch | 150 |

Table 3: Experimental settings for Contrastive $\lambda$-Repformer.

For Narrative Representation Module in $\lambda$-Representation Encoder, we used the following prompt to generate descriptions: "Give a clear, comprehensive and detailed description of the state of the objects shown in this image. For each object, mention their colors, sizes, shapes, how they are placed (upright, etc.), position within the image and relative position to other objects. Begin with the phrase 'In the image,'. Only use information that can be gained from the image. Mention the objects that appear in the sentence string below. If the objects in the sentence string are not present in the image, mention that they are not present. Sentence string: 'instruction' ." Here, we inserted the instruction sentence for each episode into 'instruction'.

### 3.4 Baselines

For comparative experiments, five baseline methods were used. We used the following experimental settings for each baseline. For each multimodal large language model (MLLM)-based method: InstructBLIP [7], Gemini [6], GPT-4V [5], we tested more than ten prompts and adopted the one with the best results.

**UNITER-base/large [59].** We performed fine-tuning according to the hyperparameter settings described in [59].

**InstructBLIP.** InstructBLIP assumes a single image as the image input. Therefore, we concatenated $x_{\text{before}}$ and $x_{\text{after}}$ as shown in Fig. 6, handling them as a single input image. The prompt used is as follows: "These two images show the robot executing the instruction 'instruction'. Based on them, please predict whether the robot has successfully completed the task and answer with 'success' or 'failure'." Here, we inserted the instruction sentence for each episode into 'instruction'. This approach was applied similarly across all MLLM-based model prompts.

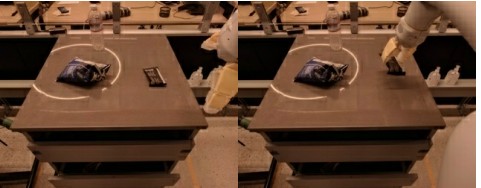

Figure 6: An example of the image input to InstructBLIP. The left and right parts show the images before and after manipulation, respectively.

**Gemini.** Gemini is capable of handling multiple images as input [6]. Therefore, during inference, we provided $x_{\text{before}}$, $x_{\text{after}}$, and the following prompt as input: "These images show the robot executing the instruction 'instruction'. The first image shows the scene before the object manipulation by the robot and the second image shows the scene after. Based on the two images and the instruction, determine whether the robot has successfully completed the task and answer with 'true' or 'false'." When we used a few-shot prompt, the model was also provided with three positive and three negative samples from the training split of the SP-RT-1 dataset, along with the sample to be evaluated. The instruction-based prompt given to Gemini was "These images show the robot executing an instruction. The first image shows the scene before the object manipulation by the robot and the second image shows the scene after. Based on the two images and the instruction, deter-

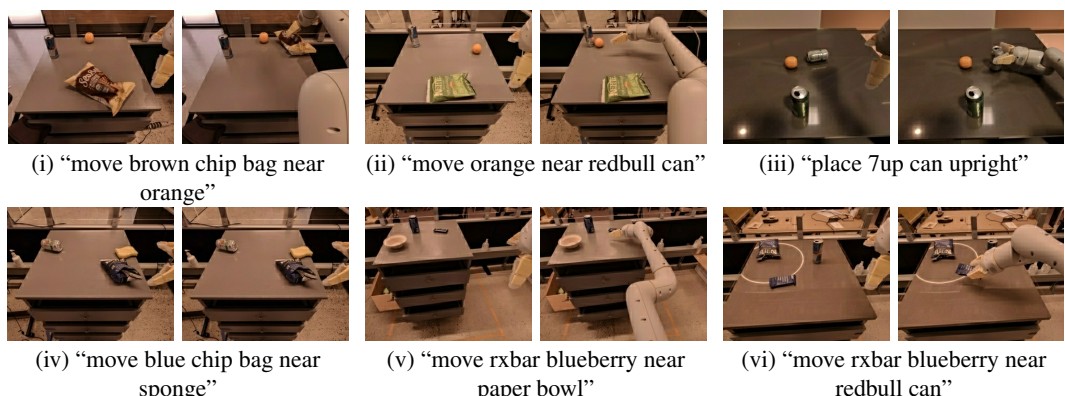

Figure 7: Samples used for the prompt in the few-shot prompted foundation model methods. The instructions given are shown below the image pairs. (i)-(iii) are positive samples, and (iv)-(vi) are negative samples.

| Model | Freeze | SR CLIP [8] | SR ViT [56] | SR DINOv2 [58] | AR | NR | Accuracy [%] |
|-------|--------|-------------|-------------|----------------|----|----|--------------|
| (i)   | ✓      | ✓           | ✓           | ✓              | ✓  | ✓  | **80.8**     |
| (ii)  |        | ✓           | ✓           | ✓              | ✓  | ✓  | 79.2         |
| (iii) |        |             |             |                | ✓  | ✓  | 73.7         |
| (iv)  |        | ✓           |             |                | ✓  | ✓  | 67.7         |
| (v)   |        |             | ✓           |                | ✓  | ✓  | 77.5         |
| (vi)  |        |             |             | ✓              | ✓  | ✓  | 79.9         |
| (vii) |        | ✓           | ✓           | ✓              |    | ✓  | 77.1         |
| (viii)|        | ✓           | ✓           | ✓              | ✓  |    | 75.2         |

Table 4: Quantitative results of the experiments where the parameters of the backbone networks were unfrozen on the SP-RT-1 dataset. Bold indicates the accuracy with the highest value. In this table, freeze, SR, AR, and NR represent the freezing of the parameters in the backbone networks, Scene Representation, Align Representation, and Narrative Representation, respectively.

mine whether the robot has successfully completed the task and answer with only 'true' or 'false'." The samples provided to the MLLMs are shown in Fig. 7 with its instruction. The samples were randomly selected.

**GPT-4V.** Similarly, GPT-4V can also process multiple images [5]. Thus, in the experiments, we inputted $x_{before}$, $x_{after}$, and the following prompt: "These images, taken from a single viewpoint camera, show the robot executing the instruction 'instruction'. Based on these images and the instruction, please determine whether the robot has successfully completed the task and answer with 'true' or 'false'." When using a few-shot prompt, as with the prompt to Gemini, we provided the model with the text prompt, three positive samples, and three negative samples. Here, the samples provided were the same as those given to Gemini. The instruction-based prompt given to GPT-4V was "Two images, taken from a single viewpoint camera, show the robot executing an instruction. Based on the images and the instruction, please determine whether the robot has successfully completed the task and answer with 'true' or 'false'."

## 4 Additional Ablation Study

### 4.1 Unfreezing Backbone Networks' Parameters

We conducted additional ablation studies where the parameters of the backbone networks were unfrozen. Table 4 shows the quantitative results. As shown in the table, the scores for unfreezing models on the RT-1 dataset were lower compared to the score for Model (i) where every backbone networks was used and frozen. On the other hand, when the backbone network was unfrozen, Model (vi) performed 0.7 points better than Model (ii). This indicates that in comparisons between models with unfrozen backbone network parameters, simpler architectures can sometimes be more effective.

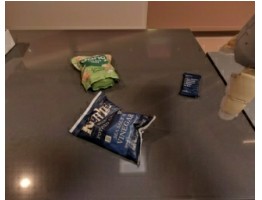 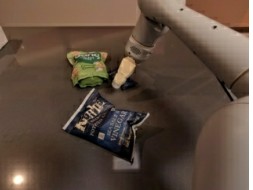

"move rxbar blueberry near blue chip bag"

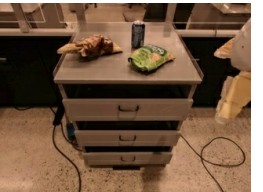 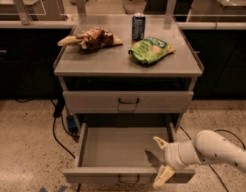

"open middle drawer"

Figure 8: A sample of Ambiguous Instruction. In this case, the given instruction was "move rxbar blueberry near blue chip bag." The ground truth label was false. The success or failure of the manipulation depends on the definition of 'near.'

Figure 9: An example of a sample in the Multimodal Language Comprehension Error category. The instruction for this sample was "open middle drawer."

However, it was shown that the proposed model, which freezes the backbone network parameters and utilizes all backbone networks, achieved the best performance.

## 4.2 Attention Mechanism

We used cross-attention instead of contrastive loss for before-after image differentiation, because cross-attention would better capture the differences required for task success prediction than other approaches such as contrastive loss. While the contrastive loss is beneficial in determining if there is a difference between features, we hypothesized that it is difficult to perform task success prediction using contrastive loss. This is because a difference between the images does not necessarily indicate task success. An example of a case where such a model could struggle is when there are slight object movements or a non-target object is moved. As a matter of fact, the cross-attention mechanism is successfully applied to image difference captioning tasks [50].

We conducted an additional ablation study to investigate the contribution of the cross-attention operation in Contrastive $\lambda$-Representation Decoder. Table 5 presents the results.

| Model | Attention Mechanism | Accuracy [%] |
|---|---|---|
| (i) | Self-Attention | $78.88 \pm 1.05$ |
| (ii) | Cross-Attention | $\mathbf{80.80} \pm 0.86$ |

Table 5: Results of additional ablation study. Bold indicates the highest value.

In this experiment, we changed the cross-attention operation to a self-attention operation to investigate its contributions. From the table, it can be observed that the accuracy of Model (i) was 78.88%, which was 1.92 points lower than that of Model (ii). This indicates that the cross-attention operation is suitable for identifying the differences between images.

## 5 Error Analysis

The confusion matrix for Contrastive $\lambda$-Repformer on the test set of the SP-RT-1 dataset includes 431, 114, 386, and 69 samples that are true positive, false positive, true negative, and false negative cases, respectively.

Thus, there were a total of 183 samples where the proposed method failed on the test set of the SP-RT-1 dataset. Table 6 shows the results of the error analysis, where we randomly selected 100 samples of failed cases. We classified them into the following six categories:

**Multimodal Language Comprehension Error:** This refers to cases where the model incorrectly interpreted visual information and instruction sentences, such as misunderstanding the target object and misinterpretation of referring expressions.

| Error type | #Errors |
|---|---|
| Multimodal Language Comprehension Error | 63 |
| Partial Visibility | 14 |
| Narrative Deficiency | 11 |
| Ambiguous Instruction | 8 |
| Erroneous Data Sample | 4 |
| Total | 100 |

Table 6: Error analysis on failure cases.

**Partial Visibility:** This category includes cases where the target object or area is only partially visible, making it difficult to make

appropriate predictions. This can occur when the target object is more than half occluded by the manipulator or other objects, or when more than half of the target object is outside the photographed scene.

**Narrative Deficiency:** This addresses cases in which the narrative from the MLLM is missing.

**Ambiguous Instruction:** This involves cases where interpretations of success or failure may vary depending on the criteria for success. Fig. 8 shows a sample included in this category. In this example, the instruction given was "move rxbar blueberry near blue chip bag." As shown in the figure, the 'rxbar blueberry' moved closer to the 'blue chip bag' before and after the object manipulation. However, the ground truth label for this example was false. In this case, the success or failure of the task depends on the definition of 'near.'

**Erroneous Data Sample:** This category covers cases where the input images of the sample are inadequate for the SPOM task, making it difficult to perform the task. For instance, a case where the instruction given is "pick a green can" and the manipulator is already grasping a green can in the $x_{\mathrm{before}}$ applies to this category.

As shown in Table 6, the main bottleneck was the Multimodal Language Comprehension Error. This issue is mainly due to the fact that the MLLM in the Narrative Representation Module generated incorrect sentences that could directly affect the success of the SPOM task. Fig. 9 shows a sample categorized as a Multimodal Language Comprehension Error. The left and right image in Fig. 9 show $x_{\mathrm{before}}$ and $x_{\mathrm{after}}$, respectively. The captions created by the MLLM for $x_{\mathrm{before}}$ was "In the image, there is an open middle drawer on a metal table. Inside the drawer, there are two objects: a sandwich and a can of soda. The sandwich is upright, while the can of soda is on its side." The captions for $x_{\mathrm{after}}$ was "In the image, there is an open middle drawer with a robotic arm reaching into it. The robotic arm appears to be picking up something from the drawer. Additionally, there is a can of soda sitting on top of the drawer." The former caption states that the middle drawer was already open before the manipulation. This makes it difficult for the model to make appropriate predictions based on the information.

This issue may be due to the difficulty of designing prompts for large language models (LLMs). Despite experimenting with many prompts and selecting the best one, erroneous generations still occurred. Indeed, object hallucination is a known challenge in image captioning by LLMs [77]. Therefore, a possible solution could be to investigate prompt designs that reduce the likelihood of such errors. For example, instead of describing everything at once, several elements could defined in advance and short responses could be obtained for each of them.

## 6  Additional Qualitative Results

Figs. 10 and 11 provide additional success examples of Contrastive $\lambda$-Repformer on the SP-RT-1 dataset and in the zero-shot transfer experiment, respectively. For the sample shown in Fig. 10 (iii), all baseline methods except InstructBLIP [7] made incorrect predictions. Likewise, for the sample displayed in Fig. 10 (ix), all baseline methods except UNITER-base [59] made incorrect predictions. It was found that for episodes with only a subtle difference between the images before and after the manipulation, the baseline methods had difficulty in making accurate predictions, whereas Contrastive $\lambda$-Repformer was able to predict appropriately.

Furthermore, all MLLM-based methods except Gemini [6] made incorrect predictions for Fig. 10 (ii), and all MLLM-based methods made incorrect predictions for Fig. 11 (ii). This indicates that even MLLM-based methods can struggle with referring expression comprehension and aligning images with natural language. From the examples in Figs. 10 and 11, it can be said that Contrastive $\lambda$-Repformer performed successfully in scenarios involving complex relational and spatial instructions, as well as in non-tabletop rearrangement settings. It can also accurately identify failures when the changes in the target object do not match the changes specified in the instructions. Especially, Fig. 10 (iv) and Fig. 11 (x), (xi), (xii) show that Contrastive $\lambda$-Repformer performed successfully in scenarios involving complex relational and spatial instructions.

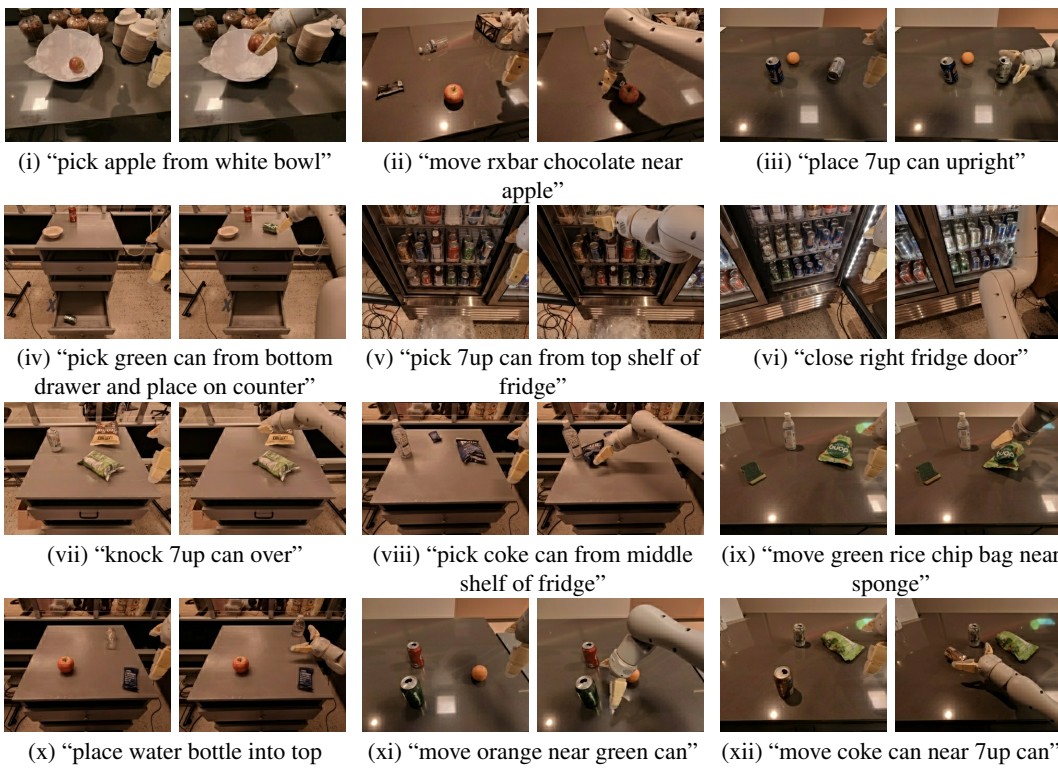

(i) "pick apple from white bowl" (ii) "move rxbar chocolate near apple" (iii) "place 7up can upright"

(iv) "pick green can from bottom drawer and place on counter" (v) "pick 7up can from top shelf of fridge" (vi) "close right fridge door"

(vii) "knock 7up can over" (viii) "pick coke can from middle shelf of fridge" (ix) "move green rice chip bag near sponge"

(x) "place water bottle into top drawer" (xi) "move orange near green can" (xii) "move coke can near 7up can"

Figure 10: Additional qualitative results on the SP-RT-1 dataset. In this figure, (i)-(vi) represent true positive cases, and (vii)-(xii) are true negative cases. These are visualized in the similar manner to Fig. 8.

Fig. 12 shows failed cases of the proposed method. Fig. 12 (i) and (ii) show the failed examples on the SP-RT-1 dataset, and Fig. 12 (iii) and (iv) exhibit the failed examples in the zero-shot transfer experiment.

Fig. 12 (i) shows an example with the instruction of "open middle drawer." The ground truth label for this example was success, because the robot opened the middle drawer. Nonetheless, our method predicted that the robot failed in carrying out the instruction. This error can be explained by the fact that most of the middle drawer lies outside the photographed area, making it hard even for humans to deduce correctly.

The instruction for the instance displayed in Fig. 12 (ii) is "pick orange from white bowl" and the ground truth label was failure. This result is most likely because the bottom of the orange is still touching the other oranges. Meanwhile, all the baseline and proposed methods predicted success. This error arises from the ambiguity of the situation, where predictions would likely be divided even among humans.

Fig. 12 (iii) presents a failed example in the zero-shot transfer experiment. In this example, the instruction sentence was "move the mug near the spam can." This sample was labeled success, whereas Contrastive $\lambda$-Repformer predicted this sample as failure. To predict appropriately, the model needs to appropriately understand both the 'mug' and the 'spam can.' In particular, to understand 'spam,' approaches such as optical character recognition are required, which makes it challenging.

Finally, Fig. 12 (iv) exhibits a failed case with the instruction of "move the apple close to the red can." Contrastive $\lambda$-Repformer predicted that the manipulator succeeded in following the instruction, while the ground truth label was failure. In this sample, there are three red objects: an apple, a red can, and a red mug. The manipulator brought the apple close to the red mug. Therefore, it is possible that the model judged the success of the manipulation based solely on the characteristic of being 'red.'

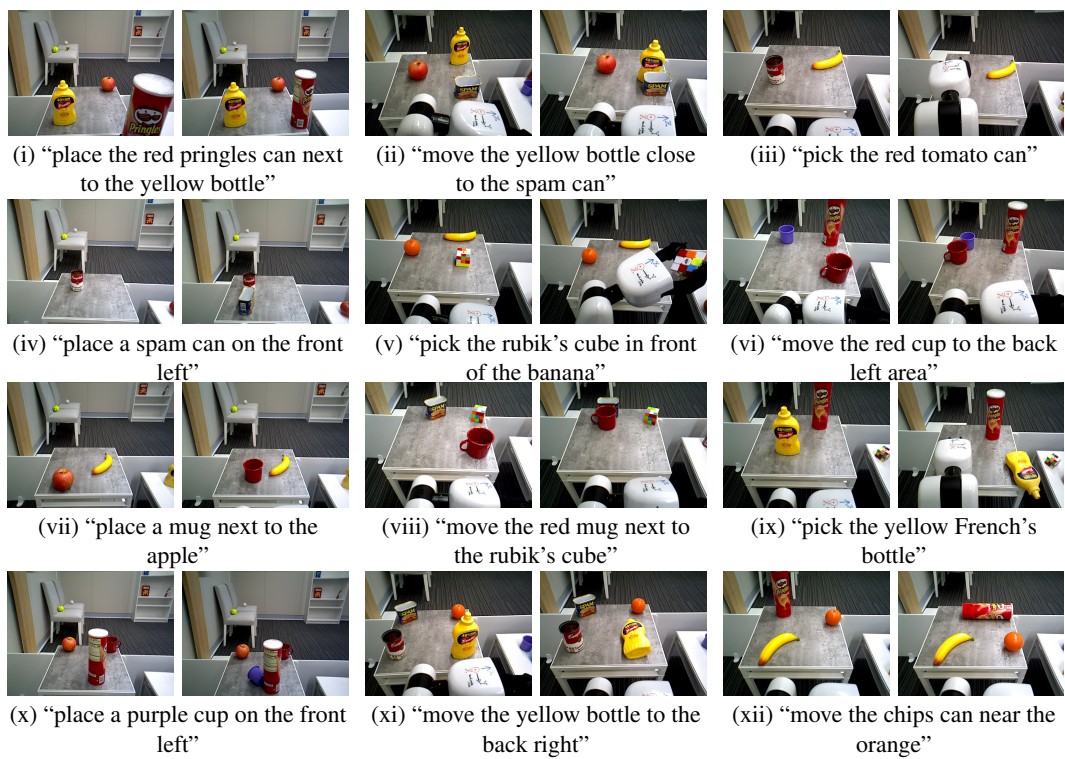

(i) "place the red pringles can next to the yellow bottle"

(ii) "move the yellow bottle close to the spam can"

(iii) "pick the red tomato can"

(iv) "place a spam can on the front left"

(v) "pick the rubik's cube in front of the banana"

(vi) "move the red cup to the back left area"

(vii) "place a mug next to the apple"

(viii) "move the red mug next to the rubik's cube"

(ix) "pick the yellow French's bottle"

(x) "place a purple cup on the front left"

(xi) "move the yellow bottle to the back right"

(xii) "move the chips can near the orange"

Figure 11: Successful examples of Contrastive $\lambda$-Repformer in the zero-shot transfer experiments. In this figure, examples (i)-(vi) show true positive cases, and (vii)-(xii) depict true negative cases. The examples are visualized in the same manner as Fig. 8.

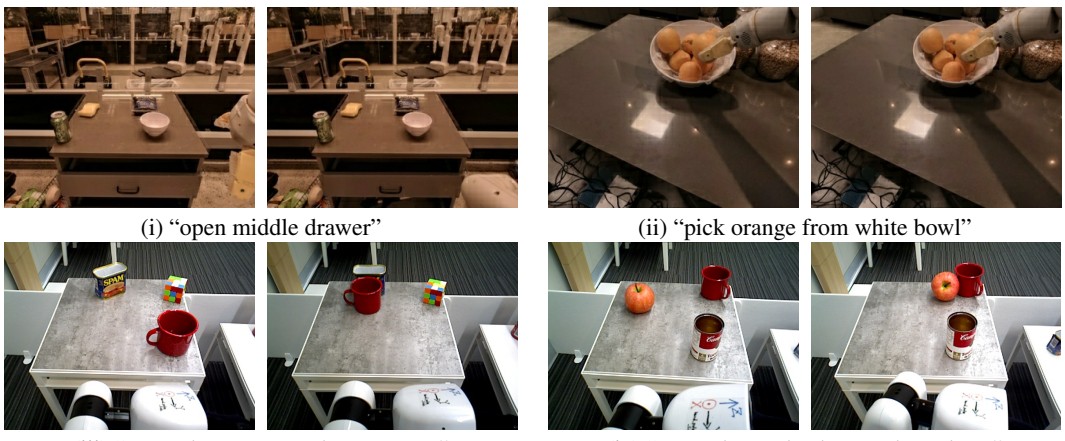

(i) "open middle drawer"

(ii) "pick orange from white bowl"

(iii) "move the mug near the spam can"

(iv) "move the apple close to the red can"

Figure 12: Failed cases of the proposed method. These are visualized in the same manner as Fig. 8.

# 7 Human Errors in Subject Experiment

Fig. 13 depicts examples where the human predictions were incorrect. In Fig. 13 (i), the instruction sentence for this sample was "pick 7up can from bottom shelf of fridge." Although the ground truth for this sample was success, the human prediction was failure. In this example, it is difficult to identify the label of the can that the manipulator grasped, as well as to determine where the can was retrieved from.

In Fig. 13 (ii), "pick the red mug" was the instruction. In this example, the mug was successfully grasped by the manipulator. However, the mug was mostly occluded, making it difficult to judge. As shown in the example, the SPOM task can be difficult even for humans.

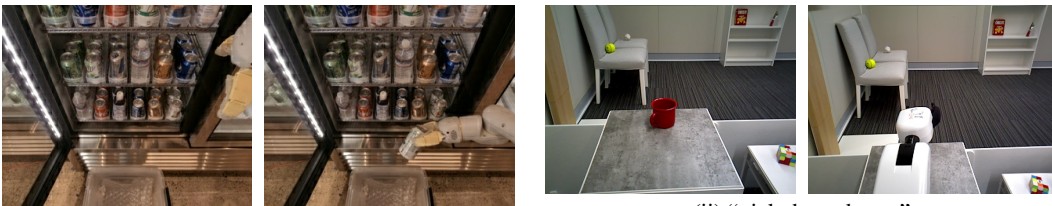

(i) "pick 7up can from bottom shelf of fridge"

(ii) "pick the red mug"

Figure 13: Samples of human errors. These are visualized in the same way in Fig. 8.

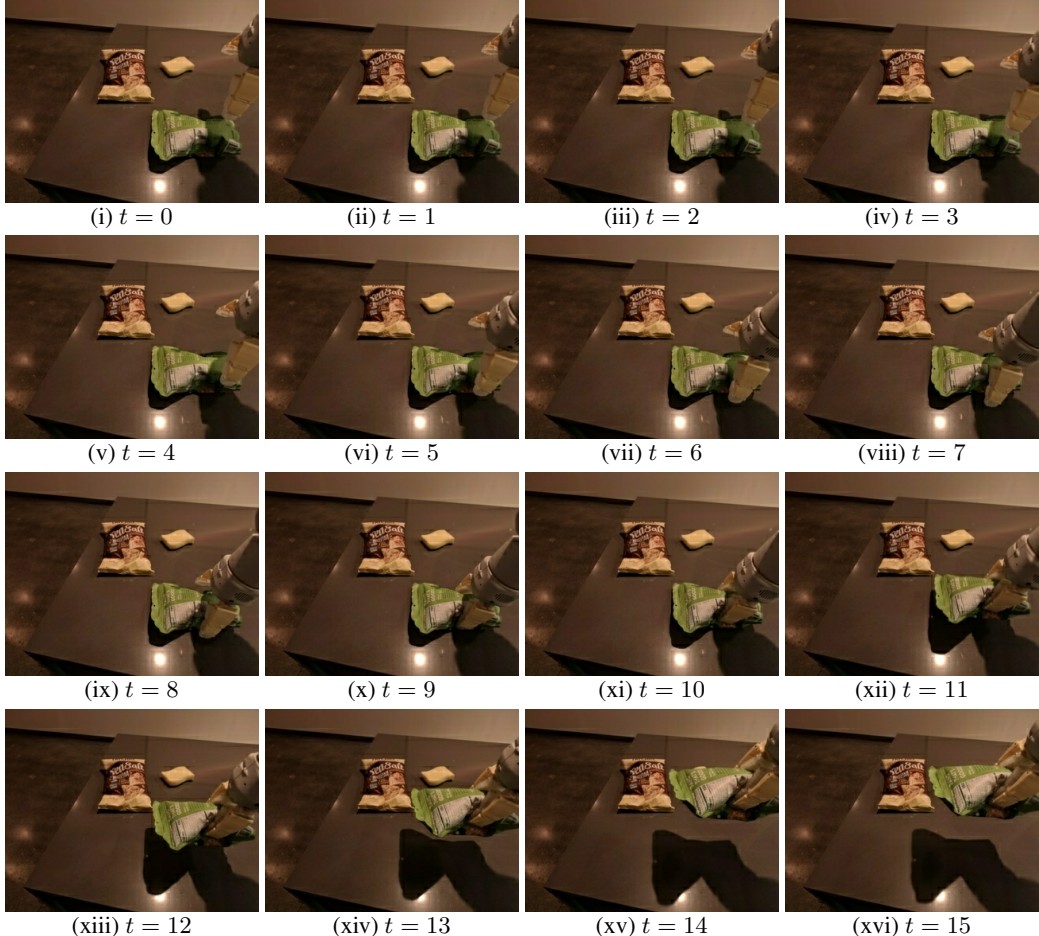

(i) $t = 0$

(ii) $t = 1$

(iii) $t = 2$

(iv) $t = 3$

(v) $t = 4$

(vi) $t = 5$

(vii) $t = 6$

(viii) $t = 7$

(ix) $t = 8$

(x) $t = 9$

(xi) $t = 10$

(xii) $t = 11$

(xiii) $t = 12$

(xiv) $t = 13$

(xv) $t = 14$

(xvi) $t = 15$

Figure 14: Successful example of a video classification problem using Contrastive $\lambda$-Repformer. The instruction was "pick green rice chip bag." The images are frames 0 to 15, as indicated by the numbers in the image. These frames are from an episode in the SP-RT-1 dataset. The instruction given to the manipulator was "pick green rice chip bag," and the ground truth label was 'success.'

## 8   Application on Video Classification Problem

We applied Contrastive $\lambda$-Repformer to the video classification problem. While the method only uses two images to perform the SPOM task, it is possible to perform video classification using it. The problem can be solved by predicting the success or failure of object manipulation at each time for the input image pairs, as follows: $(t = 0, t = 1), (t = 0, t = 2), \ldots, (t = 0, t = N-1), (t = 0, t = N)$. Here, $(t = 0, t = n)$ represents an image pair consisting of frames at times $t = 0$ and $t = n$. In this approach, video classification can be done by making predictions based on whether the proposed method outputs 'Success' at any point or continues to output 'Failure' until the end.

Fig. 14 shows a successful sample. In this sample, the instruction and ground truth label were "pick green rice chip bag" and success, respectively. The example contained 16 frames, with the success state changing at $t = 14$. The proposed method was able to detect this change appropriately. This

indicates that Contrastive $\lambda$-Repformer can also solve video classification problems. An advantage of this method is its ability to perform success prediction in real-time, unlike methods which require video input.

