# OpenReview forum: "Task Success Prediction for Open-Vocabulary Manipulation Based on Multi-Level Aligned Representations"
_robot-learning.org/CoRL/2024/Conference — CoRL 2024_

### Official Review · Reviewer_ELhE · 2024-07-03
**Good impression, needs some clarification on ablations and baselines**

**Originality:** 2
**Technical Quality:** 2
**Clarity Of Presentation:** 3
**Potential Impact:** 2
**Recommendation:** 3
**Confidence:** 4

**Review:**

This paper develops a method to do success detection given an ensemble of models using vision input (ViT), language input (BERT), or both (CLIP, InstructBLIP). While it's nice to combine lots of pretrained models together and have a small adapter layer, I believe this system for success detection may be unnecessarily complex. If I understand correctly, every one of these specialist models are frozen. I'd believe a simple unfreezing would probably yield higher performance.

The ablation experiments were well executed. I think one further level of ablations would be to have only SR, only AR, and only NR experiments to also have an understanding of how each of them alone might contribute to solving the success detection task.

For the results in Table 1 (Section 4.2), it is a reasonable start to do the zero-shot eval using the MLLMs. However, it's not exactly a fair comparison since you are capturing the success detection task in your weights, where these did not. If you used in-context learning to give the foundation models an example of a 2-3 positive/negative tasks, and maybe even provided an instruction-based prompt [1], you might end up getting a higher eval number from a foundation model.

The authors mention the challenge of occlusion that make it difficult to know if an object in the final frame is in the correct configuration. Perhaps a reasonable solution to this may be to not only use the first and last frame, but to treat this as a video classification problem.

Success detection should be a relatively simple problem. It's effectively a binary classification problem, and there should be more than enough signal in a ViT and simple language embedding to capture that task fully. For lightweight models such as BC-Z, we can do policy execution which is more complicated that success detection. Prior works using VLMs, such as PaLM-E [2] achieved 91% on RT-1 success detection, so I'm a bit surprised that the best achievable performance is only 80% here.

[1] How to Prompt Your Robot: A PromptBook for Manipulation Skills with Code as Policies - https://openreview.net/forum?id=T8AiZj1QdN

[2] PaLM-E: An Embodied Multimodal Language Model - https://palm-e.github.io/

**Quality Of The Limitations Section:**

1

**Questions For Rebuttal:**

- Why was the ViT/CLIP frozen? Could you do an experiment with only the Scene Representation, but allow gradients to flow back into the ViT? Effectively, your scene representation is frozen, but you could actually tune the scene representation to be more aware of robotics scenarios.
- It's actually quite heavyweight to have an ensemble of 5 models (ViT, DINOv2, CLIP, InstructBLIP, BERT). It would be nice to take a step back and see if a simpler architecture could implicitly capture the scene representation, aligned representation, and narrative representation jointly in the weights of the backbone.
- Could you add a baseline using some few-shot prompted foundation models (GPT, Gemini) to see if that can lift performance on the SPOM task? Would be nice to have an in-context learning baseline.
- Would it be possible to add ablations with only SR, AR, and NR? It would be interesting to understand which representation alone yields the highest accuracy.

**Robotics Focus:**

4

**Summary Of Paper:**

This paper proposes a method to do success detection for manipulation tasks by crafting an ensemble of vision and language models.

**Summary Of Recommendation:**

More experiments could be done to improve the success detection performance as well as to compare against a more competitive baseline (by using in-context learning to steer the foundation models).

---

### Official Review · Reviewer_cPTW · 2024-07-20
**Weak Accept**

**Originality:** 3
**Technical Quality:** 3
**Clarity Of Presentation:** 3
**Potential Impact:** 2
**Recommendation:** 3
**Confidence:** 3

**Review:**

Strengths:
- This is a very compelling problem and well-motivated research question
- Evaluation on standardized open-source benchmarks like RT-1 is good
- The paper is generally easy to follow and well-written
- The proposed methodology is well-motivated, design choices are explained well. I like the idea of leveraging multiple different kinds of embeddings we can get from foundation models to do success/failure detection.

Weaknesses:
- Generally, I think the kinds of failures/successes this work considers are very simple (pick/place of wrong object, high-level planning failures). It is difficult to see how it would really scale to many of the kinds of failures that show up in real-world manipulation, like imprecise alignment, or low-level failures where the arm is mostly doing the right thing but slightly oriented badly, etc.
- It would be much more interesting to close the loop -- i.e. assuming your method works well for success/failure detection, can you show real-world results where you detect failures and implement some recovery behaviors for simple tasks, even if they are hardcoded? Or, can you use this success/failure detection in a sparse reward RL setting, in sim or real? My main concern also is that "success" is a very coarse-grained definition here. There are many cases, even in RT-1, where the robot is making substantial progress in the task but fails at the last minute. For example, in the reorientation task discussed, the robot may rotate the bottle upright but then be too high up from the table surface and just drop it and topple it at the last second. I think these kinds of failure cases are much more interesting than trying to infer whether a robot picked the wrong object, etc.
- All of the baselines considered are quite simple/strawman, one is a human oracle, most of the others are just zero-shot models (GPT-4V, Gemini, InstructBLIP) for success/failure detection which are known to be quite bad at this kind of reasoning in the first place. I understand that there are some fundamental differences between REFLECT and this work, but it would really be more compelling to compare against some other kind of work like REFLECT or DROC, if you can reproduce results on one of their evaluation suites.

**Quality Of The Limitations Section:**

3

**Questions For Rebuttal:**

- Can you show some more qualitative examples of what correctly detected failed vs. successful states look like in these benchmarks? How did you get ground truth success/failure labels?
- Can you show some demonstration in the real world where you use this approach either for detecting & recovering from failures, or use the success/fail detector in some other way (i.e. sparse reward RL)?
- Is it possible to compare against some baseline that is not just a few-shot VLM?

**Robotics Focus:**

3

**Summary Of Paper:**

This work proposes a method for performing success/failure detection in robotic manipulation tasks, by ensembling several representations from multimodal foundation models.

**Summary Of Recommendation:**

Overall, the proposed idea and methodology seem sound, but I don't have enough info to contextualize how this will work in a real-world setting and how it will compare to other more compelling baselines.

---

### Official Review · Reviewer_yUdH · 2024-07-21

**Originality:** 4
**Technical Quality:** 5
**Clarity Of Presentation:** 5
**Potential Impact:** 4
**Recommendation:** 4
**Confidence:** 4

**Review:**

Strengths:
- Paper is well written and easy to follow
- The SPOM problem statement is clearly written and set up. The SP-RT-1 dataset is a practical and useful tool to ease research along this direction.
- Methodology is laid out clearly and experiments are set up in a reproducible way
- Useful ablation study across representation types and attention types
- I like the multi-level alignment architecture proposed in this paper. It is much simpler than the bottlenecked multi-level transformers needed for backbones, also leading to impressive inference time (1.6ms/sample). This highlights multiplicative gains achievable by using backbone features in a well-thought out manner.

Weaknesses:
- While the domain chosen is examined thoroughly, the method is still only applied and tested within the narrow range of tabletop manipulation. This raises concerns around the impact of this method to wider domains where mobile manipulation and non-tabletop rearrangements may occur. It is non-trivial to extrapolate given results due to multiple confounding variables.
- The paper will benefit from properly motivating its backbone choices. Also recommend the authors to look at DINO features due to their unique ability "...to contain explicit information about the semantic layout of an image" [1]
- Paper flow will benefit from better call-outs to appendix for training information, dataset information, etc.
- Paper will benefit from analysis/intuition as to why/how cross-attention based difference computation performs against contrastive loss based before-after differentiation

General Note:
I like that the paper sets up a well-scoped problem and the tools required to study it. I would like to see further experiments analyzing backbone choices and contemporary MLLMs more deeply especially to ascertain the gaps in current literature. The SPOM problem statement has great impact potential but the community needs a better understanding of gaps given the haze of MLLMs and backbone choices available. The authors are well placed to continue deeper along this direction.

[1] "Vision Transformers Need Registers", Timothée Darcet, Maxime Oquab, Julien Mairal, Piotr Bojanowski

**Quality Of The Limitations Section:**

3

**Questions For Rebuttal:**

- Why did you choose ViT, DINOv2 and CLIP specifically?
- Why use intermediate CLIP features for Scene Representation vector while last layer for Aligned Representation?
- Why use cross-attention instead of contrastive loss for before-after image differentiation? Could you please outline your motivation.

**Robotics Focus:**

4

**Summary Of Paper:**

The paper proposes a model called Contrastive lambda-Repformer, which curates aggregate features from various VLMs and LLMs and fuses them in a novel multi-level architecture to predict task success given a pair of before-after pictures and task in natural language. The experiments find that this model, trained on subset of RT-1 dataset, does better than contemporary MLLMs on predicting task success for SP-RT-1 test dataset. Qualitatively, in a zero-shot setting, we see similar improvements for this model against MLLMs. The paper also contributes refined SP-RT-1 dataset as part of the SPOM problem it formulates.

**Summary Of Recommendation:**

I find this paper well-written, with well founded and articulated contributions. It is sufficiently rigorous in its experiment to motivate my vote of acceptance. The paper can benefit from going deeper into this topic and reviewer has outlined potential improvements that they would like to see. While I have concern about general applicability of this exact same setup, I find the scalable nature of methodology and results convincing to warrant sharing with the community and critical debate to push along this problem axis.

---

### Author Rebuttal · Authors · 2024-08-12

We thank the reviewers and Area Chair for evaluating our submission and for their valuable comments.
We have attached the revised manuscript as a ZIP file.
In the revised manuscript, the updated parts are highlighted in blue.
We believe that all of the reviewers' comments have been addressed in the revised version of the paper.
In the revised version, we have added and analyzed additional experiments requested by the reviewers, demonstrating the contribution of our approach both quantitatively and qualitatively.

We address the motivation for the backbone architecture, additional qualitative results, generalization of failures to real-world scenarios, integration into closed-loop recovery, and comparisons with strong general baselines beyond zero or few-shot vision-language models.

---

### Decision · Program_Chairs · 2024-09-04

**Decision:**

Accept

**Comment:**

Strengths:
- Well written
- Reproducible methodology and useful ablations

Weaknesses:
- Narrow evaluation:
	- How do the failures discussed generalize to real world scenarios
	- How can the approach be integrated into a closed-loop recovery
	- Strong (general) baselines considered but not directly in domain
		- Comparison beyond zero or few-shot VLMs
- Writing:
	- Motivation of backbone
		- Choice/comparison of visual backbone
		- Cross-attention vs contrastive loss
		- Expensive ensemble
		- Decisions on what to freeze
	- Additional qualitative results and recovery

------
Updated:
Thank you to the authors for answering all the raised questions and providing additional results/updates. I appreciate the incorporation of many of these points into the manuscript which clearly explain these decisions, results, and (in some cases) references to the appendix.  The resulting paper will be substantially more impactful than the submission.  There are a few points not included and the new document is longer so please take care when editing not to weaken the presentation.